# ON THE LONG RANGE ABILITIES OF TRANSFORMERS

## ABSTRACT

Despite their dominance in modern DL and, especially, NLP domains, transformer architectures exhibit sub-optimal performance on long-range tasks compared to recent layers that are specifically designed for this purpose. In this work, drawing inspiration from key attributes of long-range layers, such as state-space layers, linear RNN layers, and global convolution layers, we demonstrate that minimal modifications to the transformer architecture can significantly enhance performance on the Long Range Arena (LRA) benchmark, thus narrowing the gap with these specialized layers. We identify that two key principles for long-range tasks are (i) incorporating an inductive bias towards smoothness, and (ii) locality. As we show, integrating these ideas into the attention mechanism improves results with a negligible amount of additional computation and without any additional trainable parameters. Our experiments also shed light on the reasons for the inferior performance of transformers on long-range tasks and identify critical properties that are essential for successfully capturing long-range dependencies. Our code is attached as supplementary.

## 1 INTRODUCTION

Enhancing the long-range capabilities of deep learning models is a central challenge for the field. This aspect is crucial for real-world time-series analysis, and can significantly boost performance in processing long-form data modalities, such as text, speech, or videos. The problem of capturing long-range dependencies encapsulates two aspects: effectiveness and efficiency. Researchers have proposed several transformer variants with sub-quadratic complexity to solve the efficiency problem. However, such mechanisms may not be beneficial without solving the effectiveness problem.

Therefore, our work focuses on the effectiveness part, which is a critical bottleneck that has been identified by Mehta et al. (2022) who observe that transformers struggle to exploit long context, and Xiong et al. (2021), which show that full-length transformers often perform comparably to local-attention-based transformers on long range tasks.

The lack of effectiveness of transformers in this setting was also exposed by the Long Range Arena (LRA) benchmark (Tay et al., 2020). This benchmark highlights that standard sequence models, such as transformers, perform poorly even on seemingly simple long-range tasks. As modern deep learning heavily relies on transformers, understanding why transformers do not perform well on these tasks, or how to improve those abilities is an essential research topic.

Motivated by recent advances in deep long sequence modeling, we delve into the question of why long-range layers such as state-space layers (Gu et al., 2021b;a; Gupta et al., 2022a) and long convolutions (Li et al., 2022; Fu et al., 2023) perform well on the LRA benchmark and other long-range tasks. We discern two simple yet significant conditions (i) an exponential decaying positional structure, and (ii) a regularized smooth global operator. Building upon these two principles, we introduce Local and Smooth Attention (LaS-Attention), a variant of attention that adheres to this pair of principles. Empirical analysis shows that this layer can boost Transformer performance on long-range tasks and narrow the gap with state-space layers and long convolutions, with negligible additional complexity compared to vanilla transformers.

**Our main contributions** encompass the following main aspects: (i) We furnish insights about long-range sequence modeling and identify the desired properties for achieving success in long-range tasks, (ii) We demonstrate that a smoothness-promoting inductive bias and positional locality are vital principles for capturing long-range dependencies. Moreover, we empirically identify that

these concepts present a crucial bottleneck affecting the long-range capabilities of current transformers. (iii) We present a novel variant of attention that is empirically proven to be effective in capturing long-range dependencies, and furthermore (iv) present an LaS-chunk variation that satisfies both the effectiveness and efficiency criteria, by having a linear complexity while maintaining high accuracy compared to other transformer variants. Finally, (v) We provide the first layer that does not rely on 1-D long convolution and yet achieves an average score higher than 70 on the LRA benchmark. This is compared to 55 of the original transformer and other transformer variants such as (Kitaev et al., 2020; Wang et al., 2020; Choromanski et al., 2020). While the new layer is not SOTA on all LRA benchmarks, the gained insights and the design elements open the door for other transformer-based long-range attention variants.

We note that our results are counter-intuitive at first, since locality and long-range are often viewed as opposing concepts. Nevertheless, this is explained by the fact that long-range layers capture far-away dependencies through a hierarchical combination of local dependencies. Such hierarchical dependencies are challenging to capture via pairwise interactions, without introducing locality.

## 2 BACKGROUND

**Global convolution layers**  Standard convolution layers are a fundamental building block of DL (LeCun et al., 1998; Ronneberger et al., 2015). These layers parameterize filter of size L and C channels with L*C parameters, where each element is defined explicitly. An emerging approach implicitly defines the convolution kernel via a learnable function (Romero et al., 2021). Namely, the kernel $k_i^h$ (filter) at position $i$ and channel $h$ is defined by a function $f^h$ such that $f^h(i) = k_i$.

These methods have three main advantages: (i) These layers can operate over an unrestricted context, as opposed to fixed-size explicit filters. (ii) The layers have sub-quadratic time dependency on sequence length, and (iii) As the number of parameters is decoupled from the sequence length, these kernels are regularized by design, which appears to be necessary for their effectiveness.

S4 (Gu et al., 2021a) and state-space layers (Gu et al., 2021b) were the pioneers to show the effectiveness of this approach, by parameterizing convolution kernels via the linear state-space model (SSM), which was then simplified using diagonal and real SSMs (Gupta et al., 2022a;b). Similar approaches by Ma et al. (2022); Lutati et al. (2023), use learnable components, including EMA and IIR filters, instead of SSMs to formulate the parameterization. As an alternative, Hyena (Nguyen et al., 2023) and CkConv (Romero et al., 2021) established the parameterization by applying standard Feedforward neural network (FFN) layers that operate on positional encoding. These approaches provide superior performance in several areas, such as NLP (Mehta et al., 2022; Wang et al., 2022; Dao et al., 2022b), speech (Saon et al., 2023a), RL (Lu et al., 2023; David et al., 2022), time series analysis, and more, especially in tasks that require capturing long-range dependencies.

**Long range transformers**  Transformers (Vaswani et al., 2017) have emerged as highly effective models for NLP Devlin et al. (2018); Radford et al. (2019), Computer Vision Dosovitskiy et al. (2020), Audio modeling, and many other tasks. However, their widespread adoption has been challenged by the quadratic cost of the self-attention mechanism and the demonstrated poor performance on long-range tasks. Many approaches have been applied to overcome this challenge and to create efficient transformer architectures (Fournier et al., 2021; Tay et al., 2022).

From the perspective of efficiency, techniques such as sparse attention (Child et al., 2019), low-rank attention (Wang et al., 2020; Winata et al., 2020), kernel-based attention (Choromanski et al., 2020), recurrent mechanisms (Hutchins et al., 2022; Dai et al., 2019), and efficient IO-awareness-based implementation (Dao et al., 2022a) proved efficient. From the perspective of effectiveness, Yu et al. (2023); Ivgi et al. (2023) combine local and global attention models hierarchically, enhancing the model's ability to handle extensive context. Zhou et al. (2022) expands long-range capabilities by applying attention in the frequency domain. Finally, (Gupta & Berant, 2020; Al Adel, 2022; Al Adel & Burtsev, 2021) employ global memory-based Attention. A recent strategy to enhance the effectivness of transformers in long-range tasks involves incorporating global convolution layers into the transformer architecture Ma et al. (2022); Saon et al. (2023b); Fathullah et al. (2023)

Alibi Press et al. (2021) is a method that enhances length extrapolation in transformers by adding a positional-based linear bias to the attention scores. It computes attention as follows:

$$D_L := \begin{bmatrix} 0 & 0 & \cdots & 0 \\ 1 & 0 & \cdots & 0 \\ 2 & 1 & \cdots & 0 \\ \vdots & \vdots & \ddots & \vdots \\ (L-1) & (L-2) & \cdots & 0 \end{bmatrix}, \quad \text{Attention}(Q,K,V) = \text{softmax}\left(\frac{QK^T - m \cdot D_L}{\sqrt{d_k}}\right) V \tag{1}$$

where $D_L$ is the distance matrix of size of the sequence length $L$ multiplied by the causal mask.

**The Long Range Arena (LRA) benchmark**  In recent years, numerous long-range transformer models have been introduced to address the inherent scalability and performance issues associated with long sequences in transformers. The LRA benchmark has emerged as a sought-after dataset tailored for evaluating these models across a variety of long-context scenarios, tasks, and data types. By offering a common ground for comparison, LRA scrutinizes model capabilities with sequences ranging from 1K to 16K tokens, encompassing text, visual data, and mathematical expressions. Recently, it has been shown that global convolution layers such as S4 Gu et al. (2021a) perform much better than transformers on this benchmark.

## 3  ANALYZED LONG-RANGE DEPENDENCIES

This research starts with a systematic attempt to understand the reasons behind the inferior performance of transformers in long-range modeling, compared to state-space layers and long convolution layers. We initiate our analysis by evaluating the transformer's capability to model long-range dependencies, focusing on aspects of expressiveness, optimization, and generalization, aiming to identify the core bottleneck.

The overall claim of this section is that the observed sub-optimal performance of transformers on long-range tasks does not arise necessarily from issues of optimization or expressiveness, which are inherent to the architecture. Rather, it is likely a matter of generalization, which can be mitigated effectively by incorporating appropriate inductive bias. This insight motivated our research, which explores the nature of long-range inductive bias and how it can be incorporated into transformers.

**Expressiveness**  Transformers are high-capacity models, which makes expressivity less likely to be the root cause of failure in long-range tasks. To demonstrate that expressiveness is not the root of the problem, we make two arguments: (i) We observe that when training vanilla transformers (equipped with positional encoding) on the LRA benchmarks including the validation set, large transformers can achieve near 100% accuracy, illustrating their capability to shatter the LRA benchmarks. (ii) In Theorem 1, in Appendix B we show that a single layer of a transformer (with positional encoding at the layer level) with $N$ heads and a sufficiently large hidden dimension, can express any state-space layer with $N$ channels. This can be substantiated by the fact that each channel of the state-space layer incorporates a long convolution kernel $K$, which can be expressed via the attention matrices. Note that our proof holds for any kernel $k$, not only for kernels constructed through state-space parametrization, and therefore it further elucidates the relationship between transformers and global convolution layers (see Sec. 2) by demonstrating that transformers are theoretically more expressive.

**Optimization**  Long-range dependencies are often associated with optimization issues, such as exploding and vanishing gradient problems. The following two arguments support the view that this is not the primary bottleneck in transformers: (i). Unlike RNNs, transformers do not process tokens through recurrent steps. Rather, they parallelize the processing of every pair of tokens using the self-attention mechanism, ensuring direct interaction between all pairs. Furthermore, each pair of tokens is processed in the same manner, thus there is no reason to assume that gradients are more likely to vanish or explode on long interactions than on short interactions. Moreover, in the insightful work of Orvieto et al. (2023), it was empirically demonstrated that vanishing and exploding gradient issues on the LRA benchmarks arise from a high number of non-linear operations between distant tokens, which is identified as one of the advantages of linear over standard RNNs. Similar to linear RNNs, in transformers the amount of nonlinearity is constant and does not depend on the distance between tokens. (ii). Transformers make extensive use of normalization layers, such as layer normalization and softmax, as well as residual connections, which makes them relatively stable.

Instead of a lack of expressiveness and pathological optimization dynamics, we claim that the primary factor behind the suboptimal performance of transformers on long-range tasks is probably the lack of **generalization**, caused by an unsuitable inductive bias that results in an unfavorable hypothesis class. In other words, the existing transformers overfit the long-range data. We support this claim with two observations: (i) models exhibiting exceptional performance on LRA benchmarks tend to contain layers with strong inductive bias, such as state-space layers, Exponential Moving Average (EMA), or other specialized layers Ma et al. (2022), and (ii) as shown in Sec. 5, there is a significant improvement in the performance of our models on the LRA benchmark as the amount of data increases. This does not occur as much for other transformers. This implies that with the right type of inductive bias, the model's ability to fit the underlying data distribution increases.

## 4 METHOD

We begin by exploring ways to incorporate suitable inductive bias into the transformer architecture. By observing specially designed long-range layers, we learn that exponentially decaying kernels and kernel smoothness are often promoted. We then explain how we incorporate these principles into the attention layers.

Sec. 3 presents the motivation for incorporating inductive bias to shape the hypothesis class favorably towards long-range dependencies, which can mitigate the generalization gap. However, the existence and specific characteristics of such inductive bias remain unclear. To address these questions, we aim to discern the common key principles underlying the design choices in layers that successfully capture long-range dependencies (See Appendix E for more details). Given the wide variety of long-range layers, including state-space layers (Gu et al., 2021b;a; Gupta et al., 2022a; Hasani et al., 2022; Smith et al., 2022), Toeplitz NNs Qin et al. (2023a), linear diagonal RNNs Gupta et al. (2022b); Orvieto et al. (2023), and long convolution layers Li et al. (2022); Fu et al. (2023), and the fact that these layers are built on many design principles such as unique initialization (HIPPO Gu et al. (2020)), regularized parameterization (NPLR Gu et al. (2021a), diagonal Gu et al. (2022); Gupta et al. (2022a), full kernel Fu et al. (2023)), numerically stable computation Gu et al. (2021b), and additional mechanisms such as novel gating Ma et al. (2022); Mehta et al. (2022) and normalization methods Orvieto et al. (2023), discerning the exact reasons why these layers perform well, especially when compared to Transformers, is a challenging task.

We, therefore, delve into the investigation of those layers. This discussion will be based on observing the kernels of several long-range layers, see Fig. 1.

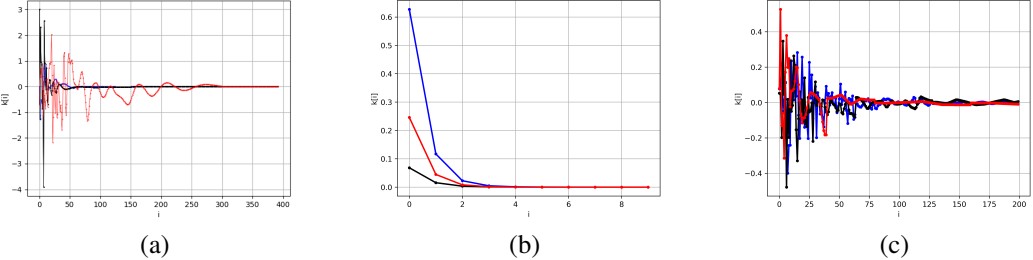

(a)        (b)        (c)

Figure 1: Examples of random kernels of several long-range layers, such as (a) S4 Gu et al. (2021a), (b) Mega Ma et al. (2022), and (c) SGConv Li et al. (2022).

A common design choice in long-range layers is to use convolution kernels with an exponential decaying structure (Li et al., 2022). This trend of exponential decay can be seen in Fig. 1. It is integrated into the kernels through initialization (Fu et al., 2023), parameterization (Gu et al., 2021a; Gupta et al., 2022a; Li et al., 2022; Ma et al., 2022), or computation (Qin et al., 2023a). All these convolutional kernels have a decaying structure, that is, the weights for interactions with closer neighbors are larger than for those with more distant ones.

Li et al. (2022) also suggest that having regularized convolutional kernels is essential for capturing long-range dependencies. Following this, Fu et al. (2023) empirically demonstrate that smoothness can be a powerful tool for kernel regularization. This smoothness can be achieved by reducing

the size of kernel weights in the time domain using a squashing operator, which enforces sparsity, leading to smoothness in the frequency domain. Moreover, layers such as EMA or linear-state space layers are naturally smooth in their design, as can be seen in Fig. 1(b).

## 4.1 LOCAL AND SMOOTH (LaS) ATTENTION

While smooth and exponentially decaying kernels are associated with a long-range inductive bias, it is unclear if such principles are pertinent to convolution kernels only or to any global operator. To further explore this matter empirically, we introduce the Local and Smooth (LaS) attention, a mechanism that modifies the attention computation by adjusting the attention matrix to incorporate a long-range inductive bias into the attention operator.

A comprehensive depiction of the LaS attention is provided in Fig. 2. The principle of smoothness is implemented by applying 1-D average pooling to each row in the attention matrix, while the exponentially decaying principle is enacted by the element-wise multiplication of the attention matrix at each head with a non-learnable locally decaying matrix. It is worth noting that, similarly to self-attention, both our local and smooth operators can manage unrestricted context with varying lengths.

Formally, one head of self-attention is given as:

$$\text{Attention}(Q, K, V) = \text{softmax}\left(\frac{QK^T}{\sqrt{d_k}}\right) V \tag{2}$$

Given this formulation, the $c$ LaS attention head can be defined by:

$$\text{LaS-Attention}_c(Q, K, V) = \text{AP}\left(\text{softmax}\left(exp(-\alpha_c D_L) \odot \left(\frac{QK^T}{\sqrt{d_k}}\right)\right)\right) V \tag{3}$$

where AP denotes an operator that executes 1-D average pooling individually for each row of the attention matrix. Given an input sequence of length $L$, the dimensions of the attention scores $\left(\frac{QK^T}{\sqrt{d_k}}\right)$ and $exp(-\alpha_c D_L)$ are $L \times L$. Average pooling is applied with corresponding padding to preserve the identical shape of the original attention scores. Lastly, our LaS attention contains two operators: the smooth operator implemented by AP, and the local operator implemented by the Exponentially Locally Decay (ELD) operator, which is defined by: ELD $: \mathbb{R}^{L \times L} \to \mathbb{R}^{L \times L}$ such that $ELD(B) = exp(-\alpha_c D_L) \odot B$, and we define the ELD matrix as $exp(-\alpha_c D_L)$. To preserve the exponential decay trend, it is essential to establish directionality within the model. Therefore, causal models are consistently used in our work. This is achieved by defining the matrix $D_L$ as the distance matrix multiplied by the causality mask. It is noteworthy that our added mechanism incurs negligible computational overhead and does not introduce any additional learnable parameters.

To control the decay rate in the c-th attention head, we utilize different values of $\alpha_c$ across various attention heads. We utilized distinct $\alpha_c$ values across the attention heads, instead of per position in the sequence to facilitate each head focusing on dependencies of a uniform scale, and provide a natural approach to operate on sequences with varying lengths. Hence, the model can capture a spectrum of local dependencies at multiple scales within each layer. This, in turn, facilitates the recognition of global dependencies at the level of the entire model, creating a hierarchical blend of local interactions that translate to global long-range dependencies.

The bottom part of the rightmost panel of Fig. 2 presents sample LDM matrices for different values of $\alpha_c$. Note that the ELD matrices $S_{i,j}$ are Toeplitz matrices, which can be succinctly represented by their first row. The first row values for different values of $\alpha_c$ are depicted in the top part of the same panel of Fig. 2. It can be observed that these Toeplitz matrices bear a resemblance to convolutional kernels, exhibiting a relatively similar structure and rule, particularly with simpler global convolution layers, such as Mega Ma et al. (2022), see Fig. 1(b).

**Initialization of** $\alpha_c$   To encourage attention heads to focus on varying types of dependencies, we regulate the effective lengths of the LDM matrices across distinct channels. This is achieved by creating a sequence of evenly spaced $\alpha_c$ values in exponential space. To facilitate a straightforward comparison with the standard transformers, we set $\alpha_0 = 0$ in the first attention head, and remove any positional decaying bias, which results in a vanilla attention head. In particular, we initialized $\alpha_c$

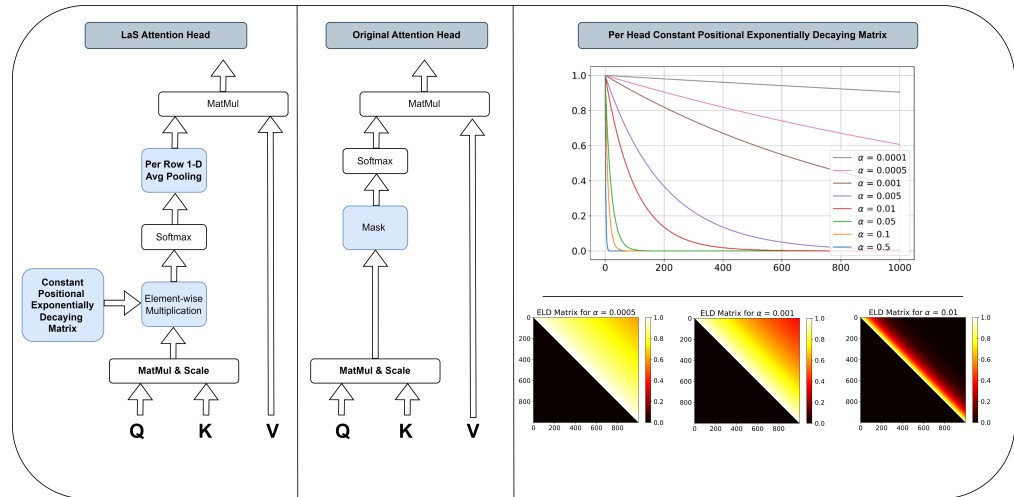

Figure 2: (Left) Our Local and Smooth (LaS) attention. (Middle) Original attention. (Right) Visualization of our local operator and ELD matrices that are discussed in Sec. 4.1

exponential-uniformly in $[0, B]$ (namely, enforce that $\exp(-\alpha_c)$ is uniformly distributed in $[0, B]$), where B is a hyper-parameter in the interval $(0, 1)$.

## 5  EXPERIMENTS

We empirically evaluate our method on standard long-range benchmarks. We first compare LaS-Attention with other transformer variants and long-range layers in Sec 5.1. Then, in Sec. 5.2, we justify our design choices by ablating each of the model's components. Finally, in Section 5.3, we investigate how various factors, such as the amount of training data and the context length of the attention influence performance. The experimental setup remains consistent across all subsections, and is described in detail in Appendix. A. Additional experiments are introduced in the appendix. For instance, the attention matrices of LaS attention are visualized in Appendix D, and the evaluation of LaS attention on NLP tasks is detailed in Appendix C.

### 5.1  RESULTS ON LONG-RANGE TASKS

In this section, we present and analyze our findings on long-range tasks, focusing on the LRA benchmark and variations of sequential MNIST.

**LRA** Tab. 1 compares the performance of our method to several previously published Transformer-based models and long-range layers. In comparison to the Transformer-based methods, including strong sub-quadratic competitors such as Reformer Kitaev et al. (2020), Linformer Wang et al. (2020), and Performer Choromanski et al. (2020), LaS Attention consistently improves performance on all the evaluated tasks, and it outperforms the previous best model Luna transformer Ma et al. (2021) in this group by a margin of 12.04%.

Specifically, on the Image task, our model outperforms all the other transformer variants by a margin of at least 22%. This finding may be attributed to the inductive bias towards smoothness and locality in our method, which is not only relevant for long-range tasks but also for natural signals, such as vectorized images.

Additionally, we propose a variant with linear complexity named LaS-Chunk attention that segments input sequences into fixed local blocks of size 128 to ensure minimal loss of local information. As can be seen, this variant surpasses all transformer methods on all tasks, with the exception of the Pathfinder task. LaS-Chunk even outperforms more computationally intensive models such as vanilla transformers, which compute the full-attention matrix and have a quadratic complexity, with an average accuracy boost of 11.34%.

Table 1: (**Long Range Arena**) Accuracy on the full suite of long range arena tasks, together with training speed and peak memory consumption comparison on the Text task with input length of 4K.

| Models | ListOps | Text | Retrieval | Image | Pathfinder | Path-X | Avg. | Speed | Mem. |
|--------|---------|------|-----------|-------|------------|--------|------|-------|------|
| *Transformers* | | | | | | | | | |
| Transformer | 36.37 | 64.27 | 57.46 | 42.44 | 71.40 | – | 54.39 | – | – |
| Local Attention | 15.82 | 52.98 | 53.39 | 41.46 | 66.63 | – | 46.06 | – | – |
| XFM‡ | 37.11 | 65.21 | 79.14 | 42.94 | 71.83 | – | 59.24 | 1× | 1× |
| Reformer | 37.27 | 56.10 | 53.40 | 38.07 | 68.50 | – | 50.67 | 0.8× | 0.24× |
| Linformer | 35.70 | 53.94 | 52.27 | 38.56 | 76.34 | – | 51.36 | 5.5× | 0.10× |
| BigBird | 36.05 | 64.02 | 59.29 | 40.83 | 74.87 | – | 55.01 | 1.1× | 0.30× |
| Performer | 18.01 | 65.40 | 53.82 | 42.77 | 77.05 | – | 51.41 | **5.7×** | **0.11×** |
| Luna-256 | 37.98 | 65.78 | 79.56 | 47.86 | 78.55 | – | 61.95 | 4.9× | 0.16× |
| **Our transformers** | | | | | | | | | |
| LAS | **53.05** | **79.28** | **85.56** | **70.44** | **81.62** | – | **73.99** | – | – |
| LAS-chunk | 46.21 | 79.11 | 83.84 | 64.90 | 54.61 | – | 65.73 | – | |
| *Models that rely on global convolutions* | | | | | | | | | |
| S4-v1 | 58.35 | 76.02 | 87.09 | 87.26 | 86.05 | 88.10 | 80.48 | – | – |
| S4-v2 | 59.60 | 86.82 | 90.90 | 88.65 | 94.20 | 96.35 | 86.09 | – | – |
| SG-Conv | 61.45 | 89.20 | 91.11 | 87.97 | 95.46 | 97.83 | 87.17 | – | – |
| LongConvs | 62.20 | 89.60 | 91.30 | 87.00 | 93.20 | 96.0 | 86.60 | – | – |
| MEGA | **63.14** | **90.43** | **91.25** | **90.44** | **96.01** | **97.98** | **88.21** | 2.9× | 0.31× |
| MEGA-chunk | 58.76 | 90.19 | 90.97 | 85.80 | 94.41 | 93.81 | 85.66 | 5.5× | 0.13× |

Compared to long-range layers incorporating global convolutions, such as MEGA Ma et al. (2022) and S4 Gu et al. (2021a), our method exhibits sub-optimal performance. This suggests that there is more to learn from these layers in terms of improving transformer architectures and understanding the shape of long-range inductive bias. A potential reason for this performance gap could be the difference in directional processing. While our models operate in a causal (unidirectional) manner, the global convolution layers in the discussed methods (with the exception of S4-v1) leverage bidirectionality. In fact, moving from unidirectional processing in S4-v1 to bidirectional processing in S4-v2 was a key upgrade, demonstrating that adopting bidirectional processing in LaS attention (which can be easily achieved at the cost of doubled complexity and computational load) could further decrease the performance gap between transformers and SOTA long-range layers.

**Sequnaital MNIST** The Sequential MNIST tasks present a challenging problem by treating 2-D images as vectors. This setup ensures that the spatial relations present in the original images are reflected as long-range dependencies in the vectorized image. Permuted MNIST is a variant where the order of pixels in each image is scrambled, intensifying the challenge and preventing models from relying on locality and periodicity. The results are presented in Tab. 2. As can be observed, LaS attention enhances performance on both tasks. For instance, on sMNIST, the local and smooth operators boost performance by 0.28%, improving the scores from 98.90% to 99.18%, while on pMNIST, performance is boosted by 0.15%, from 97.90% to 98.05%.

Table 2: Accuracy (percents) for vectorized image classification on the Sequential (sMnist) and Permuted (PMnist) MNIST. All results except LaS copied from Gu et al. (2021a)

| | sMNIST | pMNIST |
|--|--------|--------|
| *Attention-Based Models* | | |
| Transformer | 98.90 | 97.90 |
| LaS (ours) | **99.18** | **98.05** |
| *Non Attention-Based Models* | | |
| LSTM | 98.90 | 95.11 |
| **S4** | **99.63** | **98.70** |

## 5.2 Justifying Design Choices and Model Variants

Our design principles yield the following model variants: (i) the LaS-attention described in Eq. 3, along with two ablated models, namely (ii) L-attention and S-attention, each containing only the local (ELD) or the smoothing operator, respectively. To empirically delve into and understand the contributions of each component, we conducted additional experiments on the LRA benchmark. To reduce computational burdens, we employ LaS chunk attention as a baseline model, and assess the

Table 3: (**Ablations**) Evaluate the contributions of the smooth and local operators within our method by comparing the Exponentially Locally Decaying (ELD) operator with Alibi on a subset of the LRA benchmark. The baseline utilized for this comparison is a Transformer with a chunk size of 128 (cT).

| Models | ListOps | Text | Retrieval | Image | Avg. |
|---|---|---|---|---|---|
| *Non-Smooth Models* | | | | | |
| CT+ALIBI | 39.94 | 66.10 | 77.61 | 42.24 | 56.47 |
| CT+ELD (L-ATTENTION) | 41.08 | 70.65 | 81.42 | 59.48 | 63.16 |
| *Smooth Models* | | | | | |
| CT+SMOOTH (S-ATTENTION) | 46.13 | 75.86 | 80.64 | 61.38 | 66.00 |
| CT+SMOOTH+ALIBI | 40.61 | 66.35 | 81.02 | 51.68 | 59.92 |
| CT+SMOOTH+ELD (LAS-ATTENTION) | **46.21** | **79.11** | **83.84** | **64.90** | **68.52** |

models on the ListOps, Text, Retrieval, and Image tasks. We avoid conducting these experiments on the Pathfinder task, since the LaS chunk struggles to generalize in this task, making the results less informative and distinct.

As can be seen in Tab. 3, each operator contributes to the success of the method. For instance, upon removal of the Smoothness operator, LaS attention significantly outperforms L-Attention, with a difference of 0.08%, 3.25%, 3.2%, and 3.52% across the evaluated tasks. Alternatively, when the contribution of the ELD operator is removed, LaS attention surpasses the resutling S-Attention by 5.13%, 8.46%, 2.42%, and 5.42% for these tasks.

**Relation to and Differences from Alibi** Both our Exponentially Locally Decaying (ELD) operator and Alibi Press et al. (2021) manipulate the attention matrix via the distant matrix, albeit with varying motivations and impacts. Alibi was created with the intention of achieving length extrapolation, whereas our operator is designed to integrate a long-range specific inductive bias into the attention mechanism. In this light, our global operator can be seen as a type of relative positional encoding designed for long-range tasks. This distinction in motivation is manifested in the subsequent differences in the computation of the attention scores. With respect to performance and long-range capabilities, at least on the subset of the four tasks from the LRA benchmarks including ListOps, Text, Retrieval and Image tasks, Tab. 3 presents that there is a considerable margin between the methods, and it appears that the additional exponential decaying structure significantly contributes to enhancing the long-range capabilities of the model, as reflected by an average improvement of 6.69% in accuracy when not using the Smooth operator (the baseline is L-attention), and 8.61% when smoothing is added to both.

### 5.3 THE IMPACT OF DATA QUANTITY AND CONTEXT-LENGTH ON LONG RANGE TASKS

**Effective Context Length** To further understand whether our LaS transformer can capture long-range dependencies, we modified the context length within the attention layers by gradually reducing the chunk size. This modification forces the model to learn interactions up to the maximum length of the chunk size at each layer. We evaluated these models on a subset of the LRA benchmarks, including Image, Text, Listops, and Pathfinder tasks. As can be seen in Fig. 3 across all experiments, a significant decrease in performance is observed as the context window narrows. This empirical evidence indicates that our LaS attention can benefit greatly from an extended context.

**Impact of Dataset size** To delve deeper into the factors affecting the performance of our models on long-range tasks, we conducted experiments with varying amounts of training data and assessed their impact on model accuracy. Fig. 4 illustrates that as the quantity of training data increases, performance on all assessed tasks including Image, Text, Listops, and Pathfinder consistently improves. This trend suggests that with an increased volume of training data, the performance of transformers improves, potentially narrowing the performance gap with long-range layers. Moreover, this trend supports the arguments presented in Sec. 3, which proposes that the primary bottleneck for transformers on long-range tasks is the generalization gap, rather than issues related to optimization or limited expressiveness, and that this gap can be mitigated by a more suitable inductive bias.

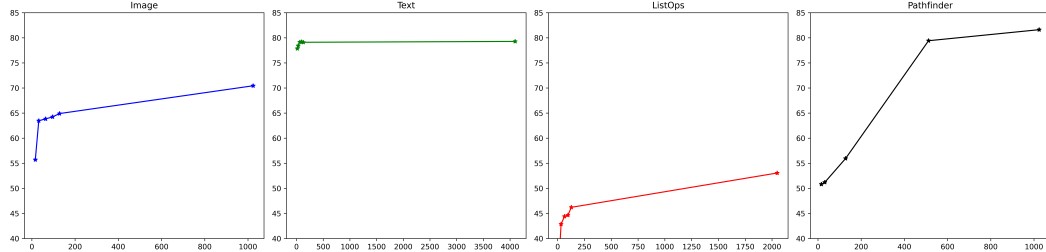

Figure 3: The effect of limiting the context window during training (x-axis) by decreasing the chunk size, and the resulting model accuracy (y-axis) on a subset of datasets from the LRA benchmark. As the context window narrows, performance decreases, highlighting that LaS attention can learn long-range dependencies at the layer level end exploit long context.

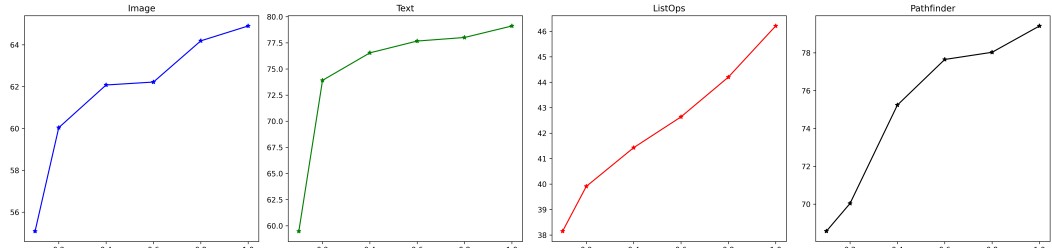

Figure 4: The impact of varying the number of training samples (x-axis) on the model accuracy (y-Axis) across a subset of datasets from the LRA benchmark.

## 6 DISCUSSION

Developing theories, tools, and well-defined concepts can significantly enhance the long-range capabilities of modern AI systems. A compelling illustration of the necessity for such tools is provided in Liu et al. (2023), which empirically demonstrates that LLMs, despite being trained on extensive data, struggle to exploit long contexts and that there are cases where performance continues to decrease as context length increases, particularly when crucial information is located in the middle of the context. Furthermore, while layers such as Mega and state-space layers achieve exceptional results on long-range tasks, it remains unclear if such layers can scale as well as transformers, and how they should be scaled up. Hence, finding alternative mechanisms and identifying the bottlenecks that prevent Transformer success in long-range tasks is an important research direction.

## 7 CONCLUSIONS

The perceived inability of transformers to learn long-range sequences has led to a proliferation of innovative methods. It is now time to examine these methods and to understand the key principles that hold transformers behind in this domain. We identify two such principles: inductive bias towards locality and smoothness along the sequence domain. Both of these are unintuitive at first, since one wishes to identify a distant signal and carry it without degradation, similarly to, e.g., the goal of the memory cells in LSTM. However, not only are these properties shared among the long-range layers, they also provide us with actionable hypotheses to verify.

Indeed, when the transformer architecture is modified such that an exponentially decaying locality kernel modulates the attention scores, the performance in long-range tasks improves. A similar improvement is obtained when an attention-smoothing term is introduced. Both modifications together bridge much of the gap in performance between transformers and the leading long-range methods. We note that while recent long-range layers have almost solved the LRA benchmark, the domain of long-range dependencies is still not understood. In this regard, this research represents an initial step in identifying and characterizing the inductive bias essential for long-range tasks, shedding light on the underlying factors required to address these dependencies.

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

## A  Experimental Setup

We conducted all our experiments using PyTorch and built our repository upon the existing S4 repository. The experiments were executed on a single V100 GPU, each running for a maximum duration of two days. To ensure consistency, each result was obtained by averaging over three different seed values. In all experiments, we employ causal transformers with 8 heads. Our training procedure and hyperparameters remained aligned with the configurations pre-specified in the S4 repository for analogous tasks. Exceptions include modifications aimed at saving computational resources, such as reducing model width, decreasing the number of epochs, and adjusting batch size, which were not optimized. The learning rate, which was determined through a grid search over the range [1e-3, 1e-4] and the orignal learning rate in the S4 repository for the corresponding task, and setting the dropout for 0 in all experiments.

The hyperparameters of the LaS attention layer are: (i) the value of B, which controls the values of $\alpha_c$, and (ii) the window size in the 1-D average pooling layer in the smooth operator, denoted by P. Hyperparameter tuning was executed via grid search on the following grid: $B \in [0.0001, 0.001]$, $Pin[3, 5]$. The final set of hyperparameters for each task is presented in Tab 4. Hyperparameters that changed from the original configuration of S4 and were optimized are denoted by (*).

Table 4: The values of the best hyperparameters found for the LRA benchmark. LR is learning rate and WD is weight decay. BN and LN refer to Batch Normalization and Layer Normalization.

|  | Depth | Features $H$ | Norm | Pre-norm | LR* | Batch Size | Epochs | WD | P* | B* |
|---|---|---|---|---|---|---|---|---|---|---|
| **ListOps** | 6 | 256 | BN | False | 1e-3 | 50 | 50 | 0.01 | 5 | 0.001 |
| **Text** | 4 | 64 | BN | True | 1e-4 | 50 | 20 | 0 | 5 | 0.0001 |
| **Retrieval** | 6 | 256 | BN | True | 0.002 | 64 | 20 | 0 | 5 | 0.001 |
| **Image** | 6 | 256 | LN | False | 1e-3 | 50 | 100 | 0.01 | 3 | 0.001 |
| **Pathfinder** | 6 | 256 | BN | True | 0.004 | 64 | 100 | 0 | 3 | 0.001 |

## B  Theorems and proofs

**Theorem 1.** *One head of self-attention can express one channel of state-space layer*

**Assumption 2.** Our assumptions are: (i) We assume that the sequence length of the input to the transformer is at most $L + 1$, and it include one additional empty token (similar to classification token) at the end. (ii) The hidden dimension of the transformer is equal to $L + 1$, implying that the key, query, and value matrices have dimensions of $(L + 1) \times (L + 1)$. (iii) The positional encoding function is an indicator function, defined as:

$$\text{PE} : \mathbb{R} \to \mathbb{R}^{L+1}$$

$$\text{PE}_j(u_i) = \begin{cases} 1 & \text{if } j = i, \\ 0 & \text{otherwise.} \end{cases}$$

and the positional encoding are concatenate to the input $u$, namely,

$$u' = \text{PE}(u) \circ u \tag{4}$$

*Proof.* We will demonstrate that under the assumptions specified in 2, Theorem 1 holds true.

We initiate our proof by revisiting the concept of state-space layers. To maintain a level of generality that is applicable to various forms of state-space layers, we consider a general state-space parameterization. The recurrent rule of such a system can be succinctly represented by a convolutional kernel in the following manner:

$$y = k * u$$

for some kernel $k$, where $k$ denotes a kernel, and $*$ represents the operation of non-circular convolution, and $u := (u_0, u_1, \cdots, u_{L-1})$ and $y := (y_0, y_1, \cdots, y_{L-1})$ are scalar sequences, namely $u, y \in \mathbb{R}^L$.

Traditionally for state-space layers, a kernel $k$ is parameterized by the system matrices $A$ and the input and output matrices $B$ and $C$, such that $k_i = f(A, B, C, i) = CA^iB$.

The applicability of our proof extends to any convolutional kernel $k := (k_1, \ldots, k_i, \ldots, k_L)$, which encompasses the kernels employed in various architectures like Hyena Poli et al. (2023), CKConv Romero et al. (2021), Focus Lutati et al. (2023), TNN Qin et al. (2023a), and others.

This convolutional kernel can be expressed via matrix multiplication $y = A_k u$ as follows:

$$
\begin{bmatrix} y_0 \\ y_1 \\ \vdots \\ \vdots \\ \vdots \\ y_{L-1} \end{bmatrix} = \begin{bmatrix} k_1 & 0 & 0 & 0 & 0 & 0 \\ k_2 & k_1 & 0 & \ddots & \ddots & 0 \\ \vdots & k_2 & k_1 & \ddots & \ddots & 0 \\ \vdots & \ddots & \ddots & \ddots & \ddots & 0 \\ k_{L-1} & \ddots & \ddots & k_2 & k_1 & 0 \\ k_L & k_{L-1} & \ldots & \ldots & k_2 & k_1 \end{bmatrix} \begin{bmatrix} u_0 \\ u_1 \\ \vdots \\ \vdots \\ \vdots \\ u_{L-1} \end{bmatrix}, \tag{5}
$$

We will demonstrate that with a given kernel $k$, it is possible to manipulate the attention mechanism, and specifically modify the keys, queries and values matrices $(W^k, W^q, W^v)$ to replicate the convolution. To do so, we assume that the input sequence $u$ include one additional empty token (similar to classification token) at the end.

The construction is outlined as follows:

- $W^v = cI_{L+1}, \quad c = \sum_{t=1}^{L} k_t$

- $W^q := W^q_{i,j} = \begin{cases} \ln A_k[i,j] & \text{if } i, j \in [L] \\ \ln c_i, \quad c_i = \sum_{t=i+1}^{L} k_t & \text{if } j = L+1, i \in [L-1] \\ \ln c & \text{if } j = L+1, i = L+1 \\ \ln 0 & \text{otherwise} \end{cases}$

$$
= \begin{bmatrix} \ln k_1 & \ln 0 & \ln 0 & \ln 0 & \ln 0 & \ln 0 & \ln c_1 \\ \ln k_2 & \ln k_1 & \ln 0 & \ddots & \ddots & \ln 0 & \ln c_2 \\ \vdots & \ln k_2 & \ln k_1 & \ddots & \ddots & \ln 0 & \ln c_3 \\ \vdots & \ddots & \ddots & \ddots & \ddots & \ln 0 & \ln c_i \\ \ln k_{L-1} & \ddots & \ddots & \ln k_2 & \ln k_1 & \ln 0 & \ln c_{L-1} \\ \ln k_L & \ln k_{L-1} & \ldots & \ldots & \ln k_2 & \ln k_1 & \ln 0 \\ \ln 0 & \ln 0 & \ldots & \ldots & \ln 0 & \ln 0 & \ln c \end{bmatrix} \tag{6}
$$

- $W^k = \sqrt{d_k} I_{L+1}$

where $I_{L+1}$ is the identity matrix of size $L + 1$. Please note that the definition of $c_i$ and $c$ enforces that the sum of the exponents be identical across the rows of $W^q$.

We begin by revisiting the formulation of self-attention:

$$
\text{Self-Attention}(u) = \text{softmax}\left( \frac{(u'W^Q)(u'W^K)^T}{\sqrt{d_k}} \right)(u'W^V) \tag{7}
$$

Based on assumption 2 (iii), and for reasons of simplicity, we assume that the input $u$ does not affect the attention matrix. Therefore $u' = PE(u)$. In practice, this can be achieved by nullifying (set to zeros) the weights associated with the input and preserving those associated with the positional encoding (PE).

$$
\text{Self-Attention}(u) = \text{softmax}\left( \frac{(\text{PE}(u)W^Q)(\text{PE}(u)W^K)^T}{\sqrt{d_k}} \right)(u'W^V) \tag{8}
$$

Note that $u' = PE(u)$ is the identity matrix $I_{L+1}$ of size $L + 1$:

$$\text{Self-Attention}(u) = \text{softmax}\left(W^Q\right)\left(uW^V\right) \tag{9}$$

Now, since $W^V$ is a scalar matrix, it commutes. By applying the definitions of the row-softmax function and the values of $W^Q$ matrix from Eq. 6, we obtain:

$$\text{Self-Attention}(u) = ZW^V u \quad = Zcu, \quad Z = \text{softmax}\left(W^Q\right) \tag{10}$$

$$\forall a, b \in [L] : Z_{a,b} = \frac{\exp(\ln(A_k[a,b]))}{c} = \frac{A_k[a,b]}{c} \rightarrow Z_{:L,:L} = A_k \frac{1}{c} \tag{11}$$

where $Z_{:L,:L}$ is the leading principal $L$-submatrix of $Z$.

By plugging Eq. 11 in Eq. 10, ignoring the output representation of the empty token, and assuming that the values corresponding to the empty token are zeros, it is simple to demonstrate that:

$$\text{Self-Attention}(u) = A_k u \tag{12}$$

This construction demonstrates that for a single channel of a state-space layer characterized by a kernel $k$, and the associated matrix $A_k$, there exist values of attention head matrices $W^q, W^v, W^q$ such that the self-attention mechanism becomes equivalent to the state-space layer. $\qquad\square$

## C  LaS ATTENTION FOR LANGUAGE MODELING

Despite LaS-Attention not being originally designed for language modeling as it relies on non-textual principles, we have evaluated our model on an NLP task to provide a more comprehensive view of the empirical capabilities of our layer. We assessed four transformer variants on the Wikitext-103 dataset for predicting the next token. Utilizing a BERT-like model architecture with 12 layers and a model width of 768, each model was trained with a context length of 512, and we employed same hyper-parameters across the experiments. We measured the perplexity for four variants: (1) vanilla attention, (2) LaS attention, and two ablations: (3) L-attention and (4) S-attention, averaging the results over two seeds. Fig. 5 presents the perplexity trends during training. It is evident that L-Attention closely matches the original model's performance, while the S-attention variants tend to fall behind. At the conclusion of training, vanilla attention achieves a perplexity of 20.20, just edging out L-attention, which sits at 20.34. S-attention and LaS attention record perplexities of 21.69 and 21.87, respectively.

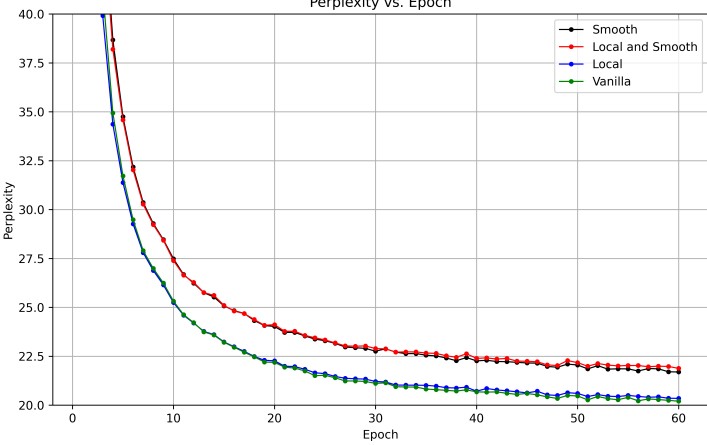

Figure 5: Evaluating LaS-Attention variants in NLP via the wikitxt-103 benchmark.

# D  VISUALIZE ATTENTION MATRICES

In Fig. 6, we present a visual analysis of attention matrices obtained from both the LaS and vanilla attention models across different layers, with both models based on a BERT-like 12-layer causal model with context length of 512, trained on Wikitext-103 for next-token prediction with the same training procedure. For clearer visualization we use min-max normalization, and we use examples from the test set of wikitext-103. As can be seen in figures, the LaS attention matrices are more attuned to long-range dependencies, especially in the upper layers, in contrast to the vanilla trans-former, which primarily focuses on short-range dependencies. Furthermore, LaS attention produces smoother attention matrices, which reduce self-attention bias toward pairwise interactions.

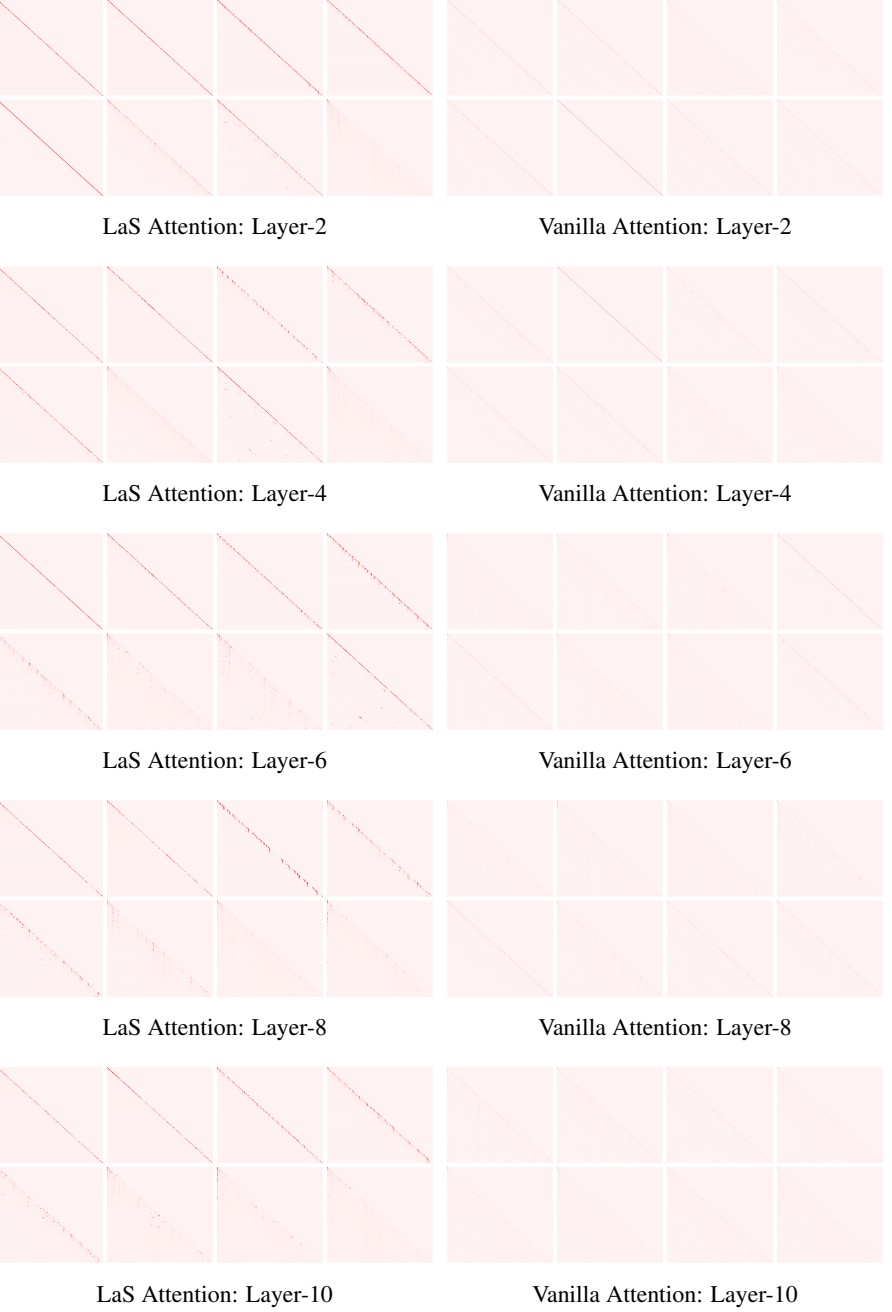

Figure 6: Visualizing the attention maps

# E  LONG RANGE LAYERS AND THEIR DESIGN ASPECTS

In Tab. 5 we provide an extensive comparison of various long-range layers and their design aspects, focusing on layers that achieve an LRA score above 0.85 or achieve new state-of-the-art results in other long-range benchmarks.

We consider multiple design aspects: (i) "Decaying structure" refers to layers that generate kernels with values decreasing over time, exemplified by state space layers which parameterize a convolutional kernel $k := (k_1, \cdots, k_L)$ such that $k_i = CA^{i-1}B$, exhibiting exponential decay where $|A| < 1$. (ii) For regularization ('R'), we identify layers that incorporate explicit regularization mechanisms vs. those layers where regularization is an inherent result of their parameterization. For example, in (Li et al., 2022) the kernels were regularized explicitly by the parameterization, and in (Fu et al., 2023) smoothness was used as a regularization tool. (iii) Unique Initialization ('U.I') can be manifested through the Hippo matrix Gu et al. (2020), used by SS, S4, DSS, and their derivatives, or through other distinctive initialization strategies. (iv) The 'Numerically Stable' designation is reserved for layers that provide explicit proof of stability or are constructed from elements specifically designed to enhance stability.

Our comprehensive analysis extends to other aspects, such as 'G' for layers relying on gating, 'C' for layers that their parameterization is based on complex numbers, and 'N' for layers explicitly employing unique normalization techniques.

Lastly, we denote layers that can be trained without recurrent rules (which are often considered a more stable approach for capturing long-range dependencies) by 'Non-Recurrent'.

Even though there are more criteria and layers Hasani et al. (2022); Zhang et al. (2023); Qin et al. (2023b), the aforementioned represent the predominant design choices.

Our review indicates that all successful long-range layers have a decaying structure, and almost none of these employ normalization or explicit regularization.

| Layer Type | Decaying structure | R | U.I | Numerically stable | G | C | N | Non recurrent |
|---|---|---|---|---|---|---|---|---|
| SS (Gu et al., 2021b) | Y | N | Y | N | N | N | N | Y |
| S4 (Gu et al., 2021a) | Y | N | Y | Y | N | Y | N | Y |
| DSS (Gupta et al., 2022a) | Y | N | N | Y | N | Y | N | Y |
| GSS (Mehta et al., 2022) | Y | N | N | Y | Y | N | N | Y |
| MEGA (Ma et al., 2022) | Y | N | Y | Y | Y | N | N | Y |
| S5 (Smith et al., 2022) | Y | N | N | Y | N | Y | N | N |
| SGCONV (Li et al., 2022) | Y | Y | Y | N | N | N | N | N |
| DLR (Gupta et al., 2022b) | Y | N | N | Y | N | Y | N | Y |
| H3 (Dao et al., 2022b) | Y | N | Y | Y | Y | Y | N | Y |
| FLASHBUTTERFLY (Fu et al., 2023) | Y | Y | Y | N | N | N | N | Y |
| LRU (Orvieto et al., 2023) | Y | N | Y | Y | N | Y | Y | N |
| TNN (Qin et al., 2023a) | Y | N | N | N | N | N | N | Y |

Table 5: Mapping of layers to their design aspects. 'R' for regularization, 'N' for normalization, 'G' for gating, 'U.I' for Unique Initialization, 'C' for parametrization over $\mathbb{C}$. We denote by 'Non Recurrent' layers that can be computed without recurrent steps.

