# OpenReview forum: "On the Long Range Abilities of Transformers"
_ICLR.cc/2024/Conference — Submitted to ICLR 2024_

### Official Review · Reviewer_nsrb · 2023-10-17

**Soundness:** 1 poor
**Presentation:** 2 fair
**Contribution:** 2 fair
**Rating:** 3
**Confidence:** 4

**Summary:**

The paper proposes an architectural modification for the transformer's attention mechanism, which allows it to generalize better to longer inputs. The authors argue that the key principles for long-range tasks are (1) inductive bias toward smoothness; and (2) locality. By modifying the transformer's attention with these principles in mind, the authors achieve empirical gains in the Long-Range Arena (LRA) benchmark.

**Strengths:**

1. Modifying the transformer architecture to better support long-range inputs is an important direction of research.
2. The empirical gains look significant, but I am not sure they were compared to the right baselines.

**Weaknesses:**

1. The paper contains many assumptions that are inaccurate or unjustified. For example:
>The lack of effectiveness of transformers in this setting [long context]

implies that transformers are ineffective, in general, on long inputs. I am not sure that this is the case - I agree that transformers could be improved, but the main problem seems to me that most transformers cannot even process long inputs. When the input *does* fit in the transformer's context window, most results that I've seen are not that bad. In other words, claiming that "transformers are ineffective in long context" requires some experiments and justification, to show exactly what the authors mean, instead of relying on one paper who said so a few years ago.

Another inaccurate and unjustified claim appears in Section 3:
>the observed sub-optimal performance of transformers on long-range tasks does not arise necessarily from issues of optimization or expressiveness, which are inherent to the architecture. Rather, it is a matter of generalization

This claim assumes that transformers are sub-optimal (compared to what?) on long-range tasks (which?), and claims that the issue is generalization. This is a bit of a vacuous claim, since "Generalization" can contain anything. If the model reaches a zero training loss but is worse at test time, is that generalization? Is that the kind of "generalization" that the authors refer to?

The paper contains further inaccurate claims like:
>Long-range dependencies are often associated with optimization issues, such as exploding and vanishing gradient problems.

which were not justified and references were not provided. I agree that in the past RNN era, this was indeed the common belief. But can this claim be said on "long-range dependencies" in general?

As a final example for inaccurate and unjustified claims, Section 3 says that:
> on the LRA benchmarks including the validation set, large transformers can achieve near 100% accuracy,

but then says that:
> the existing transformers underfit the long-range data.

Don't these two claims contradict each other? If not, there should be provided some justification, numbers, and references.

2. Motivation - the authors motivate their solution with the assumption that:
>smooth and exponentially decaying kernels are associated with a long-range inductive bias

This "axiom" isn't clear - I am not sure what exactly the authors mean by "smoothness". And in general, why? Why would *decaying* any kind of attention would improve any long-range modeling?
I was not convinced that these are indeed related to the problem/solution.

3. Evaluation - the experiments section does not provide any details regarding the underlying model - number of layers, number of parameters, whether it was pretrained, etc. I am also not sure what is the right baseline - Table 1 lists a variety of baselines, but are they comparable in terms of sizes? Which of them is the baseline that has the exact same number of parameters and layers, but without the proposed attention modification?

4. Clarity - there are many parts of the paper which are unclear. For example, the introduction says:
>We discern two simple yet significant conditions (i) an exponential decaying positional structure, and (ii) a regularized smooth global operator.

>furthermore (iv) present an SL-chunk variation

At this point in reading the paper, this is meaningless to me.
As another example, the next Background section mentions many terms, but completely meaningless for the uninformed reader. For example:

> An emerging approach implicitly defines the convolution kernel via a learnable function (Romero et al., 2021). Namely, the kernel kh
i (filter) at position i and channel h is defined by a function fh such that fh(i) = ki.

For a reader who did not read the paper by Romero et al. (2021), this doesn't mean anything.
The rest of the Background section continues to cite dozens of papers, while these citations *hurt* the readability of the paper if the reader had not read the referenced papers. Later, Alibi (Press et al., 2021)  is mentioned, along with its equations (Equation 1), without any elaboration, leaving the reader to try to understand the equations and their notations, while Alibi is completely irrelevant to the proposed approach.

Further, some figures in the paper are not explained nor elaborated. Figure 1, for example, shows "Examples of random kernels of several long-range layers", without mentioning (1) what does the x axis mean; (2) what is the y axis; (3) what are the different colored curves, and what should the reader understand from this figure.

**Questions:**

### Questions

1. What are the number of layers, number of parameters, was the model pretrained, etc.?
2. All tasks in the experimental section are somewhat synthetic. Do the authors' conclusions hold for **text**-based tasks as well, e.g., long-document summarization, long-document QA, etc?
3. Which of the baselines in Table 1 is the baseline that has the exact same number of parameters and layers, but without the proposed attention modification?
4. The models that "rely on global convolutions" in Table 1 perform significantly better than the proposed LaS attention model. If so, what is the benefit of using the proposed LaS model? Who is expected to benefit from using LaS?

### Summary

While the high-level idea is interesting, I feel like this paper suffers from too many weaknesses (detailed above). I thus recommend rejection at this time, and hope that the authors would improve writing and justify their motivation in the future.

---

> ### Author Response · Authors · 2023-11-23
> **Official Comment by Authors**
>
> Thank you for the comprehensive review and the constructive feedback.
>
> > The paper contains many assumptions that are inaccurate or unjustified
>
> We disagree that the paper contains many inaccurate or unjustified assumptions. It appears the reviewer may not be fully cognizant of recent developments in the domain, which our paper discusses in the introduction and background sections.
>
> > For example: The lack of effectiveness of transformers in this setting [long context].. implies that transformers are ineffective, in general, on long inputs. I am not sure that this is the case - I agree that transformers could be improved, but the main problem seems to me that most transformers cannot even process long inputs. When the input does fit in the transformer's context window, most results that I've seen are not that ba,  In other words, claiming that "transformers are ineffective in long context" requires some experiments and justification, to show exactly what the authors mean, instead of relying on one paper who said so a few years ago.
>
>
> The statement that the reviewer extracted from our manuscript ("transformers are ineffective in long context") is immediately substantiated in the next two paragraphs.
>
> It seems that the reviewer may not be aware of critical and well-established results in long-range modeling, which are also discussed in our introduction and background. Specifically, state-space layers such as S4 [S4] (ICLR 2022), DSS [DSS] (NeurIPS 2022), H3 [H3] (ICLR 2023), S5 (ICLR 2023), and other variants such as Gated state-space [GSS] and Liquid state-space [LSS] are known to achieve significantly better results than transformers on long-range tasks (please see full reference list in the shared response).
>
>  Notably, these layers outperform the best-known transformer variants by 20 points on the Long Range Arena Benchmark, a key benchmark in the field, and they do so with fewer parameters and even when compared to transformers trained on the entire sequence length. For more details, please refer to Table 1.
>
> Additionally, studies such as [H3, GSS] have shown that the interleaving of such layers enhances performance on long-range language modeling tasks. This trend is also evident in RL [RL1,RL2], speech processing [S1,S2,S3], and other areas [F1,HY2] (see citations in the shared response).
>
>
>  > This claim assumes that transformers are sub-optimal (compared to what?) on long-range tasks (which?), and claims that the issue is generalization. This is a bit of a vacuous claim, since "Generalization" can contain anything. If the model reaches a zero training loss but is worse at test time, is that generalization? Is that the kind of "generalization" that the authors refer to? “compared to what? ..” :
>
> Please refer to the first sentence in the section you cited: “This research starts with a systematic attempt to understand the reasons behind the inferior performance of transformers in long-range modeling, **compared to state-space layers and long convolution layers**”.
>
> *Additionally, please consider the third paragraph in the introduction:** “Motivated by recent advances in deep long sequence modeling, we delve into the question of **why long-range layers such as state-space layers** (Gu et al., 2021b;a; Gupta et al., 2022a) and long convolutions (Li et al., 2022; Fu et al., 2023) perform well on the LRA benchmark and other long-range tasks”.
>
> Moreover, the first sentence in the abstract states “.. transformer architectures exhibit sub-optimal performance on long-range tasks **compared to recent layers that are specifically designed for this purpose**. In this work, drawing inspiration from **key attributes of long-range layers, such as state-space layers, linear RNN layers, and global convolution layers”.**
>
> For more discussion regarding these questions, please see point #4 in the shared response.
>
>
> > Long-range tasks (which?) :
>
> The main focus of the paper is the Long Range Arena benchmark. However, the findings are also applicable to other long-range tasks, including long-range NLP (as seen in [H3,DSS]), RL [RL1,RL2], speech processing [S1,S2,S3], and more [F1,S4,HY2] (please see citations in the shared response).
>
> > “Generalization" can contain anything”
>
> Generalization is a well-defined term widely used in machine learning papers. It encompasses various issues such as the absence of regularization, inadequate inductive bias, or a small training dataset. It does not pertain to everything. For instance, it does not cover multiple aspects of expressiveness or optimization (despite being related).
>
> > “If the model reaches a zero training loss but is worse at test time, is that generalization?”
>
> Yes, this is an example of a lack of generalization.
>
> > Is that the kind of "generalization" that the authors refer to?
>
> Yes.
>
> > “the existing transformers underfit the long-range data”
>
> Thanks for catching this! This is a very confusing typo which is fixed in the revised manuscript.

---

> > ### Author Response · Authors · 2023-11-23
> > **Official Comment by Authors (Cont.)**
> >
> > > “Long-range dependencies are often associated with optimization issues, such as exploding and vanishing gradient problems.” which were not justified and references were not provided. I agree that in the past RNN era, this was indeed the common belief. But can this claim be said on "long-range dependencies" in general?
> >
> > We do not agree that the sentence is unjustified. For example, a few lines after the one the reviewer cited, it is written, "Moreover, in the insightful work of Orvieto et al. (2023), it was empirically demonstrated that vanishing and exploding gradient issues on the LRA benchmarks arise from a high number of non-linear operations between distant tokens".
> >
> > > Motivation - the authors motivate their solution with the assumption that “smooth and exponentially decaying kernels are associated with a long-range inductive bias”. This "axiom" isn't clear - I am not sure what exactly the authors mean by "smoothness"
> >
> >  Please refer to paragraph 4 in section 4 (method), which is one line above the line you cited: “Following this, Fu et al. (2023) **empirically demonstrate that smoothness can be a powerful tool for kernel regularization. This smoothness** can be achieved by reducing the size of kernel weights in the time domain using a squashing operator, which enforces sparsity, leading to smoothness in the frequency domain. Moreover, layers such as EMA or linear-state space layers are naturally smooth in their design, as can be seen in Fig. 1(b).”
> >
> > > Evaluation - the experiments section does not provide any details regarding the underlying model - number of layers, number of parameters,
> >
> > Please refer to the entire section on experimental setup (Appendix A), which includes details about the model, such as the number of layers, architecture, and more (table 4).
> >
> > >  Table 1 lists a variety of baselines, but are they comparable in terms of sizes? Which of them is the baseline that has the exact same number of parameters and layers, but without the proposed attention modification?
> >
> > Such tables are in a standard format and are presented similarly (without the number of parameters, FLOPs, etc.) in many key papers in the domain, such as S4, Mega, S5, and dozens more. In fact, as we mention in the experimental setup, we use models that are smaller than most of the models listed ( “Our training procedure and hyperparameters remained aligned with the configurations pre-specified in the S4 repository for analogous tasks. Exceptions include modifications aimed at saving computational resources”).
> >
> > > “furthermore (iv) present an SL-chunk variation” .. At this point in reading the paper, this is meaningless to me.
> >
> > Using chunk variants of transformers is a standard for long-range evaluations. Basically, it employs the same approach, but in chunks. We introduce the main idea a few lines before the one the reviewer cites: “Building upon these two principles, we introduce Local and Smooth Attention …”
> >
> > **Questions**
> > > What are the number of layers, number of parameters, was the model pretrained, etc.?
> >
> > The model was not pre-trained since we followed the standard protocol for the Long Range Arena benchmark. All hyperparameters are detailed in Appendix A in a dedicated table (Table 4).
> >
> > > All tasks in the experimental section are somewhat synthetic. Do the authors' conclusions hold for text-based tasks as well, e.g., long-document summarization, long-document QA, etc?
> >
> > Thank you for this question. Please refer to the new results under topic #3 in the main response. In brief, we have conducted experiments on language modeling using the Wikitext-103 benchmark. Furthermore, the LRA benchmark includes text-based tasks.
> >
> > > Which of the baselines in Table 1 is the baseline that has the exact same number of parameters and layers, but without the proposed attention modification?
> >
> > The first row in Table 1 presents the results for the baseline vanilla transformer. The size of the models based on Large-Scale (LaS) attention differs slightly, as we have adopted the hyperparameters of the S4 model as our primary baseline. It is important to note that increasing the model size is generally not known to be overly beneficial for such tasks.

---

> > > ### Author Response · Authors · 2023-11-23
> > > **Official Comment by Authors (Cont.)**
> > >
> > > > The models that "rely on global convolutions" in Table 1 perform significantly better than the proposed LaS attention model. If so, what is the benefit of using the proposed LaS model? Who is expected to benefit from using LaS?
> > >
> > > **This paper aims to provide insights about long-range modeling both with and without transformers and does not solely focus on presenting SOTA results**. To provide such insights, we investigate several key questions: (i) Why do other layers, such as state-space layers, outperform transformers in these tasks? (ii) Can transformers be enhanced by integrating techniques from those superior layers? (iii) What is the fundamental challenge in learning long-range dependencies? (we suggest it is a generalization).
> > >
> > > Additionally, there are domains in which global convolutions do not perform well or are outperformed by transformers. In such cases, our LaS approach could be beneficial.

---

### Official Review · Reviewer_tj55 · 2023-11-01

**Soundness:** 3 good
**Presentation:** 3 good
**Contribution:** 2 fair
**Rating:** 5
**Confidence:** 4

**Summary:**

The paper investigates how transformers use long-range dependencies.

The thesis of the paper is that transformers suffer from lack of generalization. They propose several inductive biases to help. Namely, smooth and exponentially decaying kernels. They are inspired by prior work in transformer variants like state-space layers.

They invent a method called Local and Smooth attention (LaS) to test their hypothesis and achieve strong results in Long Range Arena (LRA) and Sequential MNIST. The local comes from weighting with what they call Exponentially Locally Decaying (ELD). The smooth comes from a convolution operator that does average pooling.

They do various ablations to further bolster their claims about generalization and the various operations in LaS.

**Strengths:**

The empirical part of the paper is strong. The method does well and the ablations support the hypothesis. They investigate the generalization claim by manipulating context length and dataset size.

The refutation that transformers lack expressiveness or suffer from optimization problems is convincing.

Figure 2 makes the method very clear.

**Weaknesses:**

The section on identifying common design choices of prior work feels a bit handy wavy. Perhaps could be better presented in a table.

There is an error either in Equation 3 or Figure 2. They transpose the softmax and average pooling. My guess is the error is in Equation 3.

There seems to be only empirical evidence about why seemingly "unintuitive" methods work. Some theoretical justification would make the paper stronger.

**Questions:**

What is meant by "necessary conditions for achieving success in long-range tasks"? Is it meant in the strict mathematical sense? If so, I would expect to see a proof along the lines of success implies these conditions.

To further support the lack of generalization due to underfitting hypothesis, have you tried getting more data beyond what's in the LRA dataset?

Since this mechanism works in a causal manner, how well does it perform in decoding tasks like language modeling or translation?

---

> ### Author Response · Authors · 2023-11-23
> **Official Comment by Authors**
>
> Thank you for the comprehensive review and the constructive feedback.
>
> > The section on identifying common design choices of prior work feels a bit handy-wavy. Perhaps could be better presented in a table.
>
> Thank you for the suggestion. We have added a comparison table and some discussion to Appendix E (in the revised manuscript).
>
> > There is an error either in Equation 3 or Figure 2. They transpose the softmax and average pooling. My guess is the error is in Equation 3
>
> Thanks for pointing out this typo. We fixed Figure 2 in the revised manuscript.
>
> > There seems to be only empirical evidence about why seemingly "unintuitive" methods work. Some theoretical justification would make the paper stronger.
>
> Please refer to Theorem 1 in Appendix B of the revised manuscript (see point #1 in the shared response). Transformers are shown to have the capacity to express several long-range layers, such as S4. This indicates that the performance gap in such tasks is not a problem of expressiveness, but rather, a problem of generalization.
>
> Regarding the 'unintuitive' method, while we agree that smoothness and locality may seem counterintuitive, we believe that the rationale for trying such a method becomes apparent after explaining that we took inspiration from long-range layers (see Figure 1 and the first paragraphs in Section 4).
>
> Additionally, regarding why smoothness and locality enhance the model's ability to manage long-range dependencies, we hypothesize that in long-range tasks across common modalities such as images, text, and speech, long-range dependencies manifest through a hierarchical combination of local and smooth dependencies. As argued in Section 3, the difficulty in capturing these long-range dependencies stems from a generalization challenge, which can be mitigated by an appropriate inductive bias. Hence, such bias, when suitably applied, can be highly effective.
>
> > What is meant by "necessary conditions for achieving success in long-range tasks"? Is it meant in the strict mathematical sense? If so, I would expect to see a proof along the lines of success implies these conditions.
>
> In this context, 'necessary conditions' is not used as a mathematical term. Due to the confusion, we have replaced "necessary conditions" with "desired properties."
>
> > To further support the lack of generalization due to the underfitting hypothesis, have you tried getting more data beyond what's in the LRA dataset?
>
> We acknowledge that incorporating additional samples could strengthen our claims. However, we have not pursued this approach, as utilizing more data could lead to questions about the data's origin, and we aim to maintain a fair comparison with previous transformer models evaluated on the LRA benchmark (as these results have been validated many times, and thus constitute a very stable baseline).
>
> We believe that reducing the amount of data serves a similar purpose and is a much faster and more efficient method.
>
> > Since this mechanism works in a causal manner, how well does it perform in decoding tasks like language modeling or translation?
>
> Indeed, the current model is causal, but it can be adapted for non-causal tasks by converting it into a bidirectional model. This can be achieved by applying each layer twice, once in each direction, similar to other global convolution layers (like S4). Alternatively, we can adopt an approach similar to that in [A1], which extends Alibi to bidirectional by carefully defining the values in the upper part of the attention matrix.
>
> [A1] Lee, Minchul, Kijong Han, and Myeong Cheol Shin. "LittleBird: Efficient Faster & Longer Transformer for Question Answering." Proceedings of the 2022 Conference on Empirical Methods in Natural Language Processing. 2022.

---

### Official Review · Reviewer_LwQf · 2023-11-01

**Soundness:** 3 good
**Presentation:** 3 good
**Contribution:** 2 fair
**Rating:** 5
**Confidence:** 4

**Summary:**

The paper suggests modifying the attention matrix in transformer layers, drawing inspiration from models like S4, with the aim of enhancing the transformer's capacity to generalize more effectively in long-range contexts.

The authors stress the significance of two main concepts: smoothness and some kind of locality. For smoothness, they implement 1-D average pooling on every row of the attention matrix. For the locality (exponential decaying of attention), an element-wise multiplication is applied between the attention matrix at each head and a locally decaying matrix (not learnable).

LaS the, the proposed method, is evaluated on the LRA benchmark and vectorized MNIST (sequential and permuted) and compared with a few different transformer variants as well as S4, Mega, and LSTM. LaS seem to achieve the best performance among Transformers.

**Strengths:**

- The proposed method is simple and achieves better results on LRA tasks compared to other transformers.
- Ablation experiments indicate the importance of both components of LaS attention (exponential decay of attention scores and smoothness)

**Weaknesses:**

- While experiments that study the impact of context length and sample size are intriguing, interpreting the results is challenging without comparing the patterns to any other baseline. For instance, in the experiment where you restrict the context window size to examine LaS's dependence on long-range dependencies, do we have insights into how a standard transformer might be influenced by a reduced context window size?
- I believe the arguments about expressivity limitations and optimization challenges might be wrong.
   - For instance, a model can fit the training data without necessarily capturing long-range dependencies, simply by leveraging spurious features.
   - For example, I believe there are elements within the transformer block where some kind of diminishing effect might transpire as the context window becomes exceedingly long.

**Questions:**

1. Could you explain how one should read Figure 1? What is the x-axis? What is the y-axis?
2. Can you elaborate a bit more on the arguments about smoothness and its relationship to improving the model's capability to handle long-range dependencies?
3. Same question as above about locality! What's the intuition?
4. How does the Transformer architecture perform on other tasks (e.g., standard language modeling or typical image classification)? What do we sacrifice by biasing the models towards solutions that generalize better in long-range context settings?
5. Have you considered learning the biases or explored applying different patterns? How does LaS compare to a model like Synthesizer?
6. What does the final attention score matrix look like?
7. Are there any interactions or side effects from the positional encoding?

---

> ### Author Response · Authors · 2023-11-23
> **Official Comment by Authors**
>
> Thank you for the comprehensive review and the constructive feedback.
>
> > While experiments that study the impact of context length and sample size are intriguing, interpreting the results is challenging without comparing the patterns to any other baseline. For instance, in the experiment where you restrict the context window size to examine LaS's dependence on long-range dependencies, do we have insights into how a standard transformer might be influenced by a reduced context window size?
>
> The results of the full-length vanilla transformer are presented in the first line of Table 1, and they are relatively similar to those of simple local attention (which we have added to Table 1 in the revised manuscript). As can be seen, in all tasks except Pathfinder, LaS-chunk performs much better than both of the requested baselines. Therefore, we omit them from the ablations in Figures 3 and 4 for clarity. Furthermore, Figure 4 was created to strengthen the claims of Section 3, and therefore is not directly related to abating the LaS mechanism.
>
> > I believe the arguments about expressivity limitations and optimization challenges might be wrong.
> For instance, a model can fit the training data without necessarily capturing long-range dependencies, simply by leveraging spurious features.
>
> We recognize that certain edge cases may arise from the model's lack of expressiveness or optimization challenges. Examples include an inherent inability to express long-range dependencies or optimization issues such as vanishing gradients that prevent the model from learning these dependencies.
>
> However, according to the evidence provided by Theorem 1, the transformer architecture does not suffer from a lack of expressiveness, as it is more expressive than other layers that perform well on this dataset. Additionally, optimization is unlikely to be problematic since the generalization improves as the number of training samples increases (refer to Figure 4 and the detailed analysis in Section 3).
>
> Finally, please note that we have refined our claims based on this discussion (“we claim that the primary factor behind the suboptimal performance of transformers on long-range tasks is **probably** the lack of generalization”, “Rather, it is **likely** a matter of generalization”).
>
> > For example, I believe there are elements within the transformer block where some kind of diminishing effect might transpire as the context window becomes exceedingly long.
>
> Our solution shows that generalization is possible even without modifying these elements. However, it may be possible that ablating parts of the transformer would also be beneficial.
>
> > Could you explain how one should read Figure 1? What is the x-axis? What is the y-axis?
>
> Given a kernel k:= (k_1, k_2, .. k_L) of length L, axes y represent the value of the kernel (k_i) on position i. Axes x is the position i.
>
> > Can you elaborate a bit more on the arguments about smoothness and its relationship to improving the model's capability to handle long-range dependencies? Same question as above about locality! What's the intuition?
>
> First, it is important to recall that we utilize these fundamentals by drawing inspiration from global convolutional layers. For instance, Figure 1 presents relatively local and smooth kernels.
>
> Second, regarding how smoothness and locality enhance the model's ability to manage long-range dependencies, we hypothesize that in long-range tasks across common modalities such as images, text, and speech, long-range dependencies manifest through a hierarchical combination of local and smooth dependencies. Since we argue that the problem of capturing long-range dependencies reflects as a problem of generalization, such inductive bias can be very effective.
>
> > How does the Transformer architecture perform on other tasks (e.g., standard language modeling or typical image classification)? What do we sacrifice by biasing the models towards solutions that generalize better in long-range context settings?
>
> Refer to point #3 in the shared response. We evaluate LaS-attention (compared to vanilla attention and additional ablations) on standard language modeling tasks in Appendix C. It appears that the locality does not negatively impact the results, and there might be a slight improvement with further tuning of the B hyper-parameter (see the last paragraph of Section 4). However, the smoothness seems to negatively affect the perplexity, possibly reducing the transformer's ability to model complex dependencies between tokens (As the 1D Avg Pooling regularizes the bias toward the pairwise interactions).
>
> > Have you considered learning the biases or explored applying different patterns?
>
> Yes, we have considered such variants (and many more), but we didn't observe any major improvements. Please refer to our code for the exact settings and other variants at  \src\models\baselines\transformer.py, line 843:
>
> `self.alpha = alpha if not learn_local_param else torch.nn.Parameter(alpha)`

---

> > ### Author Response · Authors · 2023-11-23
> > **Official Comment by Authors (Cont.)**
> >
> > > What does the final attention score matrix look like?
> >
> > Please refer to point #2 in the shared response, as well as Appendix D. Evidently, the LaS attention matrices are better tuned to long-range dependencies, particularly in the upper layers, unlike the vanilla transformer which tends to focus on short-range dependencies. Additionally, LaS attention yields smoother attention matrices, mitigating the bias of self-attention towards pairwise interactions.
> >
> > > Are there any interactions or side effects from the positional encoding?
> >
> > Based on the experiments we conducted on Wikitext-103, it appears that our method, akin to Alibi, enhances length generalization. However, further experiments are needed to confirm this phenomenon, even though it seems plausible.

---

> > ### Comment · Reviewer_LwQf · 2023-12-01
> >
> > >The results of the full-length vanilla transformer are presented in the first line of Table 1, and they are relatively similar to those of simple local attention (which we have added to Table 1 in the revised manuscript). As can be seen, in all tasks except Pathfinder, LaS-chunk performs much better than both of the requested baselines. Therefore, we omit them from the ablations in Figures 3 and 4 for clarity. Furthermore, Figure 4 was created to strengthen the claims of Section 3, and therefore is not directly related to abating the LaS mechanism.
> > Two points:
> > 1. I don't understand in what sense the results for standard transformer and simple local attention presented in Table 1, are claimed to be similar? According to this paper, standard transformer performs better than simple local attention on all tasks.
> > 2. I thought the point about Figure 3 and 4, is not just about the performance but the pattern in which the performance changes as the size of context window or number of samples increases. So, I still think it helps if we see how this plot will look like for example for a standard transformer.
> >
> > >We recognize that certain edge cases may arise from the model's lack of expressiveness or optimization challenges. Examples include an inherent inability to express long-range dependencies or optimization issues such as vanishing gradients that prevent the model from learning these dependencies.
> > However, according to the evidence provided by Theorem 1, the transformer architecture does not suffer from a lack of expressiveness, as it is more expressive than other layers that perform well on this dataset. Additionally, optimization is unlikely to be problematic since the generalization improves as the number of training samples increases (refer to Figure 4 and the detailed analysis in Section 3).
> >
> > Generally, I find this discussion a bit inaccurate and uninformative. If the argument is to say  by adding proper inductive biases we can speed-up and facilitate the ability of transformers to learn longe range dependencies. That makes sense.
> >
> > >Refer to point #3 in the shared response. We evaluate LaS-attention (compared to vanilla attention and additional ablations) on standard language modeling tasks in Appendix C. It appears that the locality does not negatively impact the results, and there might be a slight improvement with further tuning of the B hyper-parameter (see the last paragraph of Section 4). However, the smoothness seems to negatively affect the perplexity, possibly reducing the transformer's ability to model complex dependencies between tokens (As the 1D Avg Pooling regularizes the bias toward the pairwise interactions).
> >
> > I appreciate the authors efforts for adding this additional experiments. But I am afraid this has made me a bit more confused. I would assume language modelling is a task that would benefit enhanced ability of the model to deal with long range dependency.
> >
> > >Yes, we have considered such variants (and many more), but we didn't observe any major improvements.
> >
> > I think it is worth it and it would be interesting to include the results of these experiments in the paper. that might help better justify the solution you are proposing.
> >
> > **Conclusion:**
> > Thank you for preparing the rebuttal. Considering the authors' response and the additional information provided during the rebuttal, I am still not convinced of the significance of the contributions and correctness of all claims of the paper. But I think with more rigorous experiments and tuned arguments, the paper could become much better in undressing the power and limits of transformers in learning long range dependencies.

---

### Official Review · Reviewer_hVaG · 2023-11-05

**Soundness:** 3 good
**Presentation:** 3 good
**Contribution:** 2 fair
**Rating:** 5
**Confidence:** 3

**Summary:**

The paper proposes a modification of the self-attention mechanism in Transformer architectures that facilitates the learning of long-range interactions. As global convolutional layers such as SSMs perform significantly better on long-range tasks, the authors' investigation of these models reveals that the convolution kernel of such models often have an exponentially decaying structure with additional smoothness constraints. This motivates the authors to introduce a modified attention mechanism called LaS-attention, which incorporates exponential decay and smoothing (implemented by average pool) along attention scores to incorporate these inductive biases into the architecture. In this way, the output of each attention layer is mostly only influenced by local interactions with varying degrees of locality, and long-range reasoning is captured hierarchically through compositions. This greatly increases Transformer generalization performance on the LRA benchmark and reduces the gap from global convolutional models. Ablation studies demonstrate the benefit of each proposed component. Additional experiment on sequential MNIST also show improvements with respect to this modification. To summarize the key observation, the paper claims that long-range reasoning is best implemented as compositions of mostly localized interactions as motivated by the SSMs, and demonstrated in the context of transformers.

**Strengths:**

- The writing of the paper is clear, and original as far as I know.
- The main strength of the paper is the intuition it provides; the reasoning is easy to follow and well-motivated, and it sheds light on the performance gap between SSMs and transformers on long-range tasks, which is a very important problem faced by Transformers.
- The hypothesis that long-range reasoning is best implemented as compositions of (mostly) localized interactions with exponentially decaying dependencies is well-motivated as supported by the investigation of the kernels of global convolutions, and then verified in the experiments by implementing a simple fix in the attention mechanism.

**Weaknesses:**

- Methodology: Although the insights are novel, significant, and interesting to read, the methodological novelty is limited. The proposed modification is a simple modification 1) of the pre-softmax linear attention matrix by pointwise multiplication with an exponential decay term, and then 2) smoothing of the activated attention matrix. Especially that a similar modified version appeared in previous work in (Press et al. 2021), and its main difference from 1) is effectively that the distance matrix is exponentiated.
- Experiments:  I also found the experimental aspect somewhat lacking, as only LRA and sequential MNIST is considered, where on LRA there is improvement but not up to par yet with global convolutions. It would also be interesting to know if the proposed upgrade can provide improvements on other Transformer tasks, where it already performs as SOTA.


Overall this is a very borderline paper for me as it is mainly a proof of concept, and while the insight is very valuable, I would have appreciated if the authors took the idea further.

**Questions:**

#### Question 1:
The authors claim that "It is fairly straightforward to show that a single layer of a transformer ... can express any state-space
layer" and "each channel of the state-space layer incorporates a long convolution kernel K, which can be expressed via the attention matrices"

I would like to ask whether the authors have any references for this claim or if they could provide a proof in the appendix? This seems highly non-obvious to me. I am aware of the previous work "On the Expressive Power of Self-Attention Matrices" Likhosherstov et al. 2021, which demonstrates that attention matrix can approximate sparse patterns, but I am not aware of more general results.


#### Question 2:

 In another paragraph, "the lack of generalization, caused by an unsuitable inductive bias that results in an unfavorable hypothesis class. In other words, the existing transformers underfit the long-range data."

I have two problems with this sentence:
1) is that previously the authors said that " large transformers
can achieve near 100% accuracy", which is not a sign of underfitting but overfitting;
2) is that, although I am not sure about the claim in the previous question at all, if we assume it's true, then it basically says that the issue is not a problem with the hypothesis class in the classical sense, which is all the possible expressible models (although the set of reachable models by common initialization + training procedure combinations is another question, if this is what is meant then it should be clarified).
It seems to me that the issue is either - as the authors stated - that decomposing long-range learning into a series of locality pronounced layers helps in the training procedure for generalizability, i.e. this constraint mitigates overfitting by restricting the hypothesis class rather than enlarging it. I wonder if the authors have any thoughts on this? This is an interesting question.

#### Question 3
Another question is whether the authors have any intuition about why the exponentiation of the distance matrix helps the attention compared to Alibi? This seems weird to me because applying the decay matrix before the softmax decreases large query-key dot-product values as intended, but negative dot-product values are actually increased, which actually degrades the ability of the model to reduce the dependence on certain tokens. I wonder if this is a desired effect due to some implicit regularization phenomenon of being less likely to attend to a very small select few tokens, or if this is might be harmful?

---

> ### Author Response · Authors · 2023-11-23
> **Official Comment by Authors**
>
> Thank you for the comprehensive review and the constructive feedback.
>
> >.. the methodological novelty is limited:
> Our method (Local and Smooth) is deliberately simple since we seek minimal modifications to the transformer architecture that could improve its long-range capabilities. We believe that controlled and simple experiments are essential for providing clear insights.
>
> The main novelty of the paper lies in the insights we offer. We have characterized the main challenge of the long range problem (generalization) and identified fundamental principles that make long-range layers, such as state-space layers, much better than transformers on those tasks. Please refer to “Framing of the Contribution” (point #4)  in the shared response for a more comprehensive description of the insights we provide.
>
> > Experiments: I also found the experimental aspect somewhat lacking, as only LRA and sequential MNIST is considered, where on LRA there is improvement but not up to par yet with global convolutions. It would also be interesting to know if the proposed upgrade can provide improvements on other Transformer tasks, where it already performs as SOTA.
>
> First, please see the experiment result on language modeling (point #3 in the main response, Appendix C). Second, we want to highlight that the LRA benchmark is the central benchmark in the domain.
>
> > The authors claim that "It is fairly straightforward to show that a single layer of a transformer ... can express any state-space layer" and "each channel of the state-space layer incorporates a long convolution kernel K, which can be expressed via the attention matrices" I would like to ask whether the authors have any references for this claim or if they could provide a proof in the appendix? This seems highly non-obvious to me
>
> We thank the reviewer for this comment. Our initial claim was that the kernel could easily be incorporated into key and query matrices. However, encouraged by your comment, we wrote a comprehensive formal proof, which we have included in Appendix B (Theorem 1). See point #1 in the shared response for more details. Please note that the proof holds in general settings for a large variety of global convolution layers, and it sheds additional light on the relationship between transformers and global convolution layers.
>
> >  … that previously the authors said that " large transformers can achieve near 100% accuracy", which is not a sign of underfitting but overfitting.
>
> Thank you for catching this! It was a very confusing typo that was fixed in the revised version!
>
> > Another question is whether the authors have any intuition about why the exponentiation of the distance matrix helps the attention compared to Alibi? This seems weird to me because applying the decay matrix before the softmax decreases large query-key dot-product values as intended, but negative dot-product values are actually increased, which actually degrades the ability of the model to reduce the dependence on certain tokens. I wonder if this is a desired effect due to some implicit regularization phenomenon of being less likely to attend to a very small select few tokens, or if this is might be harmful?
>
> We investigate this question through experiments. It appears that the model learns to control such phenomena by creating a non-symmetric distribution at the input to the softmax layer. Specifically, when iterating over 1,000 examples with a model trained using Large-Scale (LaS) attention, we observe that the values before the softmax layer are in the range of [-100,000,000 to 100], which could potentially handle your scenario.

---

### Author Response · Authors · 2023-11-23
**Main Response by Authors**

We thank the reviewers for their thoughtful comments and constructive feedback. Following the reviews, we have uploaded a revised version of the manuscript which contains the following material (modifications in red):

**(1) Proof that transformers are more expressive than long range layers**:

Following the review of hVaG, we prove in Appendix B (Theorem 1) that a single transformer head can express one channel of state-space layers. The proof holds in general settings and sheds more light on the relationship between transformers and global convolution layers. This proof encompasses nearly all state-space layers [S4,DSS,LSS], as well as more global convolution layers such as CKconv [CK], Hyena [HY], and more [TNN, SG, F1]). As the development of global convolution layers is a significant research area within the domain of efficient deep learning and long sequence modeling, we believe that this proof provides valuable insights to the community.

**(2) Visualize attention matrices:**

Following the review of LwQf, in Appendix D we visualize the final attention matrices obtained from LaS and vanilla attention models. As can be seen in the figures, the LaS attention matrices are more attuned to long-range dependencies, especially in the upper layers, in contrast to the vanilla transformer, which primarily focuses on short-range dependencies. Furthermore, LaS attention produces smoother attention matrices, which reduce self-attention bias toward pairwise interactions..

**(3) NLP experiments:**

Following the reviews of LwQf and nsrb,  we evaluate LaS-attentoin (compared to vanilla attention and additional ablations) on standard language modeling tasks in Appendix C. It is evident that L(local)-Attention closely matches the original model's performance, while the S(mooth)-attention variants tend to fall behind.

Finally, We would like to emphasize the paper's contribution by reframing the main problem we tackled:

**(4) Framing of the contribution:**

The seminal work of S4 demonstrated that state-space layers outperform transformers on long-range tasks, substantiated by their SOTA performance on the Long Range Arena (LRA) benchmark [LRA]. Further research, including various studies, has shown that employing global convolution layers can consistently enhance the long-range capabilities of DL models across several domains, such as NLP [H3,GSS], speech [S1,S2,S3], RL [RL1,RL2], video processing [V1], fMRI [F1], and others [HY2].

A pivotal question emerging from this significant line of research is: **What attributes make global convolution layers superior to transformers for long range tasks?** We are the first to attempt to answer this by demonstrating that global convolution layers rely on inductive bias towards smoothness and locality (as illustrated in Figure 1 and detailed in the initial paragraphs of the Methods section).

We demonstrate that such straightforward yet counterintuitive principles can enhance the long-range capabilities of transformers, particularly in the Long Range Arena benchmark, a key benchmark in this field.

Lastly, our controlled experiments offer further insights into why global convolution layers excel at long-range tasks, by highlighting that their success is rooted in simple fundamental principles that are associated with enhanced performance, extending beyond the realm of global convolution layers.

**Despite hundreds of papers designing novel and complex layers for enhanced performance on long-context tasks, the specific challenges these improvements address remain elusive. In this context, we  provide several insights:** (i) Theorem 1 demonstrates that enchanted expressiveness may not be effective. (ii) Figure 4 shows that a data-driven approach can be very effective, and (iii) Section 3 suggests that the critical barrier to the long range problem in modern DL models is probably an issue of **generalization**, rather than a problem of optimization and expressiveness. These insights advance our understanding and pave the way to new capabilities in this area.

---

> ### Author Response · Authors · 2023-11-23
> **Main Response by Authors (Cont.)**
>
> [S4] - Efficiently Modeling Long Sequences with Structured State Spaces. Albert Gu, Karan Goel, Christopher Ré. ICLR 2021.
>
> [H3] - Hungry Hungry Hippos: Towards Language Modeling with State Space Models. Daniel Y. Fu, Tri Dao, Khaled K. Saab, Armin W. Thomas, Atri Rudra, Christopher Ré. ICLR 2023
>
> [GSS] - Long Range Language Modeling via Gated State Spaces. Harsh Mehta, Ankit Gupta, Ashok Cutkosky, Behnam Neyshabur. ICLR 2022.
>
> [DSS] - Diagonal State Spaces are as Effective as Structured State Spaces. Ankit Gupta, Albert Gu, Jonathan Berant. NeurIPS 2022.
>
> [S1]- Diagonal State Space Augmented Transformers for Speech Recognition. George Saon, Ankit Gupta, Xiaodong Cui. ICASSP 2023.
>
> [S2] - It's Raw! Audio Generation with State-Space Models. Karan Goel, Albert Gu, Chris Donahue, Christopher Ré. ICML 2022.
>
> [S3]  - Structured State Space Decoder for Speech Recognition and Synthesis. Koichi Miyazaki, Masato Murata, Tomoki Koriyama. ICASSP 2023.
>
> [Rl1] - Decision S4: Efficient Sequence-Based RL via State Spaces Layers. Shmuel Bar-David, Itamar Zimerman, Eliya Nachmani, Lior Wolf. ICLR 2022.
>
> [RL2] - Structured State Space Models for In-Context Reinforcement Learning. Chris Lu, Yannick Schroecker, Albert Gu, Emilio Parisotto, Jakob Foerster, Satinder Singh, Feryal Behbahani. arXiv 2023.
>
> [V1]- Long Movie Clip Classification with State-Space Video Models. Md Mohaiminul Islam, Gedas Bertasius. ECCV 2022.
>
> [F1] - Simple Hardware-Efficient Long Convolutions for Sequence Modeling. Daniel Y. Fu, Elliot L. Epstein, Eric Nguyen, Armin W. Thomas, Michael Zhang, Tri Dao, Atri Rudra, Christopher Ré. ICML 2023.
>
> [LSS] - Liquid Structural State-Space Models. Ramin Hasani, Mathias Lechner, Tsun-Hsuan Wang, Makram Chahine, Alexander Amini, Daniela Rus. ICLR 2022.
>
> [TNN] - Toeplitz Neural Network for Sequence Modeling. Zhen Qin, Xiaodong Han, Weixuan Sun, Bowen He, Dong Li, Dongxu Li, Yuchao Dai, Lingpeng Kong, Yiran Zhong. ICLR 2023
>
> [HY] - Hyena Hierarchy: Towards Larger Convolutional Language Models. Michael Poli, Stefano Massaroli, Eric Nguyen, Daniel Y. Fu, Tri Dao, Stephen Baccus, Yoshua Bengio, Stefano Ermon, Christopher Ré. ICML 2023.
>
> [HY2] - HyenaDNA: Long-Range Genomic Sequence Modeling at Single Nucleotide Resolution Eric Nguyen, Michael Poli, Marjan Faizi, Armin Thomas, Callum Birch-Sykes, Michael Wornow, Aman Patel, Clayton Rabideau, Stefano Massaroli, Yoshua Bengio, Stefano Ermon, Stephen A. Baccus, Chris Ré. NeurIPS 2023.
>
> [CK] - CKConv: Continuous Kernel Convolution for Sequential Data. David W. Romero, Anna Kuzina, Erik J.   Bekkers, Jakub M. Tomczak, Mark Hoogendoorn. ICLR 2021.
>
> [SG] - What Makes Convolutional Models Great on Long Sequence Modeling? Yuhong Li, Tianle Cai, Yi Zhang, Deming Chen, Debadeepta Dey. ICLR 2022.
>
> [LRA] - Long Range Arena: A Benchmark for Efficient Transformers. Yi Tay, Mostafa Dehghani, Samira Abnar, Yikang Shen, Dara Bahri, Philip Pham, Jinfeng Rao, Liu Yang, Sebastian Ruder, Donald Metzler. ICLR 2020.

---

### Meta-Review · Area_Chair_8TJ8 · 2023-12-19

**Metareview:**

The paper explores improving transformers' performance in long-range tasks. Instead of totally revamping the transformer architecture like some prior works, the authors propose a some modifications incorporating smoothness and locality. Empirical studies were initially conducted on the Long Range Arena benchmark. However, the paper received underwhelming but informative response from the reviewers who pointed out several weaknesses, such as limited methodological novelty, inadequate experimental scope, and insufficient theoretical justification. We thank both the authors and reviewers for engaging during the discussion phase to improve the paper. In particular, adding the proof and additional NLP experiments were useful, but didn't resolve concerns for any of the reviewers. Nevertheless the direction of exploration by the authors can be interesting to the community, and thus we strongly encourage authors to revise the manuscript based on the reviews and submit to the next venue. Also it might be worth considering venues which emphasizes less on novelty like TMLR.

**Justification For Why Not Higher Score:**

Primarily due to concerns about the methodological novelty and the limited scope of experiments. They pointed out that the modifications to the transformer structure, while interesting, were not sufficiently novel or groundbreaking. Additionally, there was a need for more rigorous justification of the claims made in the paper. The experiments were seen as somewhat limited in scope, leading to questions about the generalizability of the results. These factors contributed to the perception that the paper's contribution was marginally below the acceptance threshold.

**Justification For Why Not Lower Score:**

N/A

---

### Decision · Program_Chairs · 2024-01-16

Reject